# Pitfalls in Diagnosing Thrombotic Thrombocytopenic Purpura in Sickle Cell Disease

**DOI:** 10.3390/jcm11226676

**Published:** 2022-11-10

**Authors:** Dimitris A. Tsitsikas, Diana Mihalca, John Hall, Jori E. May, Radhika Gangaraju, Marisa B. Marques, Marie Scully

**Affiliations:** 1Homerton Healthcare NHS Foundation Trust, London E9 6SR, UK; 2Department of Pathology, Heersink School of Medicine, University of Alabama at Birmingham, Birmingham, AL 35294, USA; 3University College Hospital, 235 Euston Road, Fitzrovia, London NW1 2BU, UK

**Keywords:** sickle cell disease, thrombotic thrombocytopenic purpura, fat embolism syndrome, therapeutic plasma exchange

## Abstract

Thrombotic thrombocytopenia purpura is characterised by microangiopathic haemolytic anaemia and red cell fragmentation on the peripheral smear, neurological involvement and thrombocytopenia. Diagnosis in the context of sickle cell disease can be challenging due to the inherent haemolytic state and the multitude of other associated complications of the latter. Specifically, fat embolism syndrome characterised by respiratory failure, neurological impairment and thrombocytopenia can be misdiagnosed this way. Confirmation of a diagnosis of thrombotic thrombocytopenic purpura requires demonstration of very low levels (<10%) of the metalloproteinase ADAMTS13 which in fat embolism syndrome is normal. Existing scoring systems used to estimate the pre-test probability for thrombotic thrombocytopenic purpura cannot be applied in patients with sickle cell disease due to the chronic underlying haemolysis. Here, we analyse the diagnostic approach in published cases of thrombotic thrombocytopenic purpura affecting patients with sickle-cell disease. The vast majority of cases were characterised by severe respiratory failure before any other manifestation, a feature of fat embolism syndrome but not of thrombotic thrombocytopenic purpura, and all received red cell transfusion prior to receiving therapeutic plasma exchange. Despite the potential overestimation of the pre-test probability using the existing scoring systems, a large number of cases still scored low. There were no cases with documented low ADAMTS13. In the majority this was not tested, while in the 3 cases that ADAMTS13 was tested, levels were normal. Our review suggests that due to many overlapping clinical and laboratory features thrombotic thrombocytopenic purpura may be erroneously diagnosed in sickle cell disease instead of other complications such as fat embolism syndrome and confirmation with ADAMTS13 testing is essential.

## 1. Introduction

Thrombotic microangiopathies (TMA) are a group of conditions defined by thrombocytopenia and microangiopathic haemolytic anaemia with resultant end organ damage. Thrombotic thrombocytopenic purpura (TTP) is a type of TMA defined by a deficiency of the metalloproteinase ADAMTS13, and can be immune-mediated (iTTP) or congenital (cTPP) [1]. Classically, the clinical presentation of TTP includes TMA features as well as neurological symptoms, renal impairment and cardiac ischemia. Different clinical and laboratory features have been used to estimate the pre-test probability of iTTP [2,3]. However, utilisation of such established scoring systems to estimate the pre-test probability of iTTP, in the context of SCD is limited by the fact that haemolysis, a fundamental characteristic of SCD even at steady state, is incorporated in both existing systems. Diagnostic confirmation requires the demonstration of severe deficiency (<10%) of ADAMTS13 and the presence of an inhibitor to ADAMTS13 [4]. The mainstay of treatment for iTTP is therapeutic plasma exchange (TPE), along with immune-modulating agents and disruption of von Willebrand factor-platelet interaction [4,5].

There is significant overlap between the clinical presentation of iTTP and complications of sickle cell disease (SCD). Like iTTP, complications of SCD include haemolysis, often with abnormal peripheral blood morphology. Fat embolism syndrome (FES) is a catastrophic complication of SCD that can cause neurological impairment, multiorgan failure, elevated lactate dehydrogenase (LDH) and thrombocytopenia similar to iTTP, making the two diagnoses challenging to distinguish [6,7,8]. FES is the result of extensive bone marrow necrosis and typically affects patients with non-homozygous SCD and those with a previously mild clinical course. Peripheral blood morphology typically shows leukoerythroblastosis with a high number of nucleated red blood cells (nRBC) [9]. FES is associated with high levels of von Willebrand factor but, unlike TTP, normal levels of ADAMTS13 [10]. Treatment of FES is red cell transfusion, but preferably, red cell exchange (RCE). Even with increasing use of RCE, FES is associated with high mortality while a large proportion of survivors are left with severe neurocognitive impairment [11]. As FES is associated with generation of high levels of proinflammatory cytokines that can cause direct tissue damage, we have previously suggested addition of TPE to RCE in an attempt to improve outcomes [11,12] and there are already a few cases treated successfully with both apheresis modalities^−^ [13,14,15]. Since ADAMTS13 activity is normal in FES, ADAMTS13 testing is the definitive method to distinguish it from iTTP.

## 2. Methods

A PubMed literature search was performed in order to identify published cases of TTP in patients with SCD and review the diagnostic approach. The terms “sickle cell” and “thrombotic thrombocytopenic purpura”/“TTP” were used. The references of identified articles as well as articles citing those selected were also searched and included herein.

## 3. Results

### 3.1. Clinical Features

Nineteen cases of iTTP in patients with SCD were identified: 9 were individual case reports [16,17,18,19,20,21,22,23,24] and a case series of 10 [25]. Eleven patients had Hb SS, 3 Hb SC and 5 Hb Sβ+. Details on the previous course of SCD was only provided in 4 cases; all had a previously mild course of the disease including 1 that was previously undiagnosed. Sixteen patients (84%) had neurological involvement either at presentation or shortly afterwards; primarily altered sensorium/drop in Glasgow Coma Scale (GCS). Fifteen (79%) patients also had severe type I respiratory failure either preceding or occurring simultaneously with the development of neurological signs. Twelve patients (63%) had varying degrees of liver dysfunction and 84% had fever at presentation or soon thereafter. There were no cases with documented cardiac dysfunction or myocardial ischaemia.

### 3.2. Laboratory/Radiological Findings

All patients had thrombocytopenia but only 7/19 (37%) had a platelet count <30 × 10^9^/L. LDH was markedly elevated in all cases with a mean of 3613 iU/L (range 280–7689). Peripheral blood morphology was variable: schistocytes were identified in all cases but, in the majority, at relatively low levels; at the same time, all had large numbers of nRBCs.

Brain imaging was documented in 7 cases: 3 computed tomography (CT) and 4 combined CT and magnetic resonance imaging. In 1 patient, intracranial haemorrhage was noted, while the remainder were normal. Chest imaging, plain radiographs and/or CT were performed in 16 cases: 2 were normal whereas 14 showed extensive infiltrates with an ARDS picture.

### 3.3. ADAMTS13 and Pre-Test Probability Estimation

As mentioned in the introduction, existing scoring systems to estimate the pre-test probability of iTTP are not suitable for application in patients with SCD due to the chronic haemolysis characteristic of the latter. In addition, several required parameters were not included in the case reports. Despite those significant limitations, at least 4/19 (21%) patients still had a score of 0 as per the French criteria and at least 11/19 (58%) had a PLASMIC score of ≤5 (Table A1). No published case documented ADAMTS13 activity of <10%; in 16 (84%) there was no mention of ADAMTS13 activity testing at all. In the 3 cases with ADAMTS13 testing, all results were normal; 2 samples were obtained before TPE and 1 after. Of interest, the only case that could potentially score 7 according to the PLASMIC scoring system was one with normal ADAMTS13 activity in a sample obtained before TPE.

### 3.4. Management and Outcomes

Eighteen patients received TPE and 1 received an infusion of donor plasma. However, prior to TPE, all patients had already received red cell transfusion (exchange or simple) with an aim to achieve Hb S levels of ≤30%. There were 2 (11%) deaths while the remainder of patients achieved complete recovery. One patient died of an intracranial haemorrhage and the other, sepsis, in the context of immunosuppression for prevention of iTTP recurrence.

## 4. Discussion

SCD is often associated with severe acute complications that share features with TTP: Neurological signs may be due to sickle-related stroke whereas overwhelming hypo splenic sepsis, a well-known complication of SCD, can cause fever, altered sensorium and thrombocytopenia if coexistent disseminated intravascular coagulation. In addition, haemolysis which is one of the hallmarks of TTP is an innate characteristic of SCD and the peripheral smear of SCD patients can be grossly abnormal thus masking or mimicking the morphology characteristic of TTP.

More than that, none of the described cases was characterised by severe renal failure requiring replacement as seen in other TMAs, e.g., haemolytic uraemic syndrome. The combination of fever, neurological signs, thrombocytopenia, and renal failure is characteristic of FES but also of TTP and can lead to confusion as the two conditions share many characteristics but also differ distinctly (Table A2). This comprehensive review of reported cases of iTTP in patients with SCD raises important questions about the potential misdiagnosis of FES as iTTP.

Collectively, reported cases demonstrated an almost universal presence of severe respiratory failure prior to the development of other symptoms or signs (including thrombocytopenia), a feature of FES but not TTP. In addition, the inconclusive morphological findings in the peripheral smear, low pre-test probability of iTTP and, most importantly, complete absence of low ADAMTS13 activity required for definitive iTTP diagnosis also bring the diagnosis to question. It should also be noted here that the markedly elevated levels of LDH observed in these reported cases is also a universal feature of FES [12].

Therefore, we suspect that many of these patients, in fact, had FES. Further supporting this possibility, we have identified 3 case reports where an original diagnosis of iTTP was revised to FES [7,8]. The positive outcomes may be explained by the fact that all patients received adequate red cell transfusion upfront, but also the possible beneficial effect of TPE in acute complications of SCD with a hyper inflammatory component such as FES.

In conclusion, the diagnosis of iTTP in patients with SCD is challenging due to many overlapping clinical and laboratory features with FES. Established scoring systems to estimate the pre-test diagnostic probability of iTTP are not relevant in SCD due to chronic hemolysis, and likely inflate the risk due to significant clinical overlap. A diagnosis of iTTP in patients with SCD, therefore, should not be made without confirmatory ADAMTS13 results. Furthermore, the possibility of alternative diagnoses should be explored swiftly.

## Data Availability

Not applicable.

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
