# Peer review of "Pitfalls in Diagnosing Thrombotic Thrombocytopenic Purpura in Sickle Cell Disease"

_jcm, 2022, doi:10.3390/jcm11226676_

Round 1

Reviewer 1 Report

I think this article provides important insights in a clinical setting often difficult to manage as regards diagnosis. Just one question: since the genesis of thrombocytopenia is different in the two conditions, do you think there are different trend in PLT decrease during TTP and FES episodes which may also help the clinician in the diagnostic algorhythm?

Author Response

Thank you for your comments. The only thing I can say about thrombocytopenia  is that  in FES it is generally more gradual, often absent at presentation, and not as severe as seen in TTP when patients present often with single figure platelets.

Reviewer 2 Report

This review of TTP cases in SCD patients outlines the difficulty of diagnosing thrombotic microangiopathies (TMA) in patients with hemoglobinopathies.

As the paper is focused on cases of so-called "TTP" published in the literature, it doesn't go in depth into the myriad other TMA syndromes, none of which are linked with ADAMTS13 deficiency. In my view the paper should remind the readers that the course of SCD can be complicated with complement-mediated TMA events, particularly in the context of delayed hemolytic transfusion reactions, probably linked to systemic complement dysregulation in these patients.

The paper should emphasize that none of the 19 patients described (if indeed this is the case) had major kidney dysfunction (e.g. needing dialysis) because it is the main difference between TTP and other TMA syndromes (so-called hemolytic uremic syndromes).

Author Response

Thank you for your review. A relevant comment has been added in the discussion section.

D. A. Tsitsikas